# Effects of Different Denaturants on Properties and Performance of Soy Protein-Based Adhesive

**DOI:** 10.3390/polym11081262

**Published:** 2019-07-30

**Authors:** Li Yue, Zhang Meng, Zhang Yi, Qiang Gao, An Mao, Jianzhang Li

**Affiliations:** 1Key Laboratory of Wood Material Science and Utilization, Beijing Forestry University, Beijing 100083, China; 2Ministry of Education, Beijing Key Laboratory of Wood Science and Engineering, College of Materials Science and Technology, Beijing Forestry University, Beijing 100083, China; 3Key Laboratory of State Forestry Administration for Silviculture of the lower Yellow River, College of Forestry, Shandong Agricultural University, Taian 271018, China

**Keywords:** soy protein, protein surface hydrophobic index, hydrophobic groups, wet shear strength, crosslinking structure

## Abstract

Chemical modification of soy protein, via crosslinking, is the preferred method for creating non-toxic, renewable, environmentally friendly wood adhesives. The denaturing process of protein is important for the adhesive performance improvement. In order to investigate the effect of different denaturing agents on the performance of soy protein-based adhesives before and after crosslinking modification. In this study, three different denaturing agents—urea (U), sodium dodecyl sulfate (SDS), and sodium hydrogen sulfite (SHS) and an epoxide crosslinking agent—Triglycidylamine (CA) were used to prepare soy protein-based adhesives. The results showed: (1) The denaturing agent unfolded protein molecules and exposed more hydrophobic groups to prevent water intrusion, which was mainly a contribution for the water resistance and performance improvement of soy protein-based adhesives. The wet shear strength was improved up to 91.3% (denaturing by urea). (2) After modifying by the crosslinking agent, the properties and performance improvement was due to the fact that the active groups on soybean protein molecules reacted with the crosslinking agent to form a crosslinking structure, and there is no obvious correlation with the hydrophobic groups of the protein. (3) The unfolded soybean protein molecules also expose hydrophilic groups, which facilitates the reaction between the crosslinking agent and protein to form a denser crosslinking structure to improve the performance of the adhesive. Particularly, after denaturing with SHS, the wet shear strength of the plywood bonded by the SPI-SHS-CA adhesive increased by 217.24%.

## 1. Introduction

The wood-based panel industry in China primarily uses formaldehyde synthetic resin adhesives and their modified products (e.g., urea-formaldehyde, phenolic, and melamine-formaldehyde resins) [1]. As such, throughout the wood-panels’ production and use life-cycle, formaldehyde and other harmful substances are released into the environment, thereby generating a pollution problem and endangering human health. In addition, aldehyde adhesives are derived from unsustainable petroleum resources [2]. Between the depletion of global petrochemical resources and the increasingly serious environmental pollution problems they cause, development of renewable and environmentally friendly wood adhesives has become an important consideration for the wood industry going forward. Protein adhesives are made from raw materials that are available in abundance [3]. Moreover, they have the advantages of being non-toxic, renewable, and environmental- friendly [4,5]; and as such, have become a research hotspot in recent years.

Soy protein is a main byproduct of agriculture [6]. Thus, it is an abundantly available, renewable, and low cost option that shows significant potential for development [7]. However, due to its low adhesion strength, poor water resistance, high viscosity, and low solid content, the soy protein adhesive has limited application in industry [8,9]. Earlier studies have attempted numerous and varied methods to chemically modify soy protein in order to improve its bonding performance—such as denaturing agent- [10,11], crosslinking- [12,13], grafting- [14,15], and nano-fiber modification [16], etc. However, the effects of different modification methods on the properties of soybean protein-based adhesives have hardly been reported. Therefore, the effects of different denaturing agents and crosslinking agents on the properties and performance of soybean protein-based adhesives were investigated. Moreover, such knowledge is useful and important for appropriate modification methods in the soy protein-based adhesive preparation and to develop more effective modification methods.

In this study, urea, SDS and SHS are common denaturants used to enhance the performance of soybean protein-based adhesives. Previous studies [17] have shown that urea mainly destroys hydrogen bonds between protein molecules, SDS as a surfactant denatures proteins, and SHS enables protein molecules to unfold by breaking disulfide bonds between molecules and hydrolyzes protein molecules [18]. Based on this, three denaturants (urea, SDS and SHS) were individually incorporated into an SPI adhesive solution in this study; which was subsequently mixed with the epoxy crosslinker (CA) to prepare three modified soy protein-based adhesives. Finally, the different denaturants’ effects on the soy protein’s surface hydrophobicity, residual rate, fracture surface micrograph and resultant adhesive performance were quantitatively assessed.

## 2. Materials and Methods

### 2.1. Materials

SPI with >85% crude protein content was purchased from Xiang Chi Grain and Oil Co., Ltd. (Shandong, China) and milled into 250-mesh flour. Urea and SHS were obtained from Lan Yi Chemical Co., (Beijing, China); and SDS was purchased from Tianjin Chemical Reagent Co., (Tianjin, China). Poplar (Populus tomentosa Carr.) veneers (40 cm × 40 cm × 1.5 mm, 8% moisture content) were purchased from the Hebei Province, China. CA was synthesized in the laboratory. Amino acid analysis was carried out using ion-exchange chromatography in an automatic amino acid analyzer (Hitachi 835-50, Japan). The SPI amino acid composition is presented in Table 1.

### 2.2. Preparation of Cross-Linker Triglycidylamine (CA)

CA was synthesized using a similar process reported in Connolly’s research [19] and the synthesis process was illustrated in Figure 1. Aqueous ammonia (A, 25 wt %) was dropwise added into an epichlorohydrin solution (E, 99.9 wt %) with a molar ratio of 4:1 (A:E) in a three-necked flask at a stirrer of 180 r/min. In addition, the mixture reacts at a temperature range from 50 to 65 °C until all ammonia was added and further reacted for 2 h at 60 °C. Since the reaction was highly exothermic, an external water-cooling circulator was required to maintain the temperature. The residual epichlorohydrin and ammonium hydroxide were removed by a vacuum distillation process at 45 °C, leaving a colorless syrup consisting of tris-(3-chloro-2-hydroxypropyl) amine. Then, a sodium hydroxide solution of 50 wt % (weight ratio of syrup:NaOH solution = 10:1) with was added to facilitate the epoxy-ring closure reaction, which took place under vigorous stirring at 50 °C for 2 h and cooled to 25 °C to obtain CA. The feather of the resultant CA was as followed: pH = 9.0, viscosity of 1200 mPa·s, epoxy value = 0.52%.

### 2.3. Preparation of Adhesive

For the SPI adhesive, 15 g of SPI was added to 85 g of deionized water and stirred for 10 min at 25 °C in order to form a homogeneous system. 1 g of denaturation agent (either urea, SDS, or SHS) was mixed with the SPI adhesive and stirred for an additional 20 min at 25 °C to develop a series of adhesives. As a control, the SPI adhesive without denaturant was also stirred at the subsequent 20 min. The resulting SPI adhesives were labeled “SPI”, “SPI-Urea”, “SPI-SDS”, and “SPI-SHS” to reflect the different denaturing agents. 5 g of CA were then mixed with each SPI adhesive, which were stirred for an additional 5 min at 25 °C to develop a second series of adhesives. The resulting adhesives were labeled “SPI-CA”, “SPI-Urea-CA”, “SPI-SDS-CA”, and “SPI-SHS-CA”. The adhesive formulations are shown in Table 2.

### 2.4. Surface Hydrophobicity Measurement

The adhesive samples were placed in an oven at 120 ± 2 °C to cure completely then ground into ~200 mesh powder in ceramic mortar to measure the surface hydrophobic index, using an analytical method reported by Mozafarpour et al. [20]. Sodium 8-anilino-1-naphthalenesulfonate (ANS) was used as the hydrophobic fluorescent probe. At room temperature, the soybean protein samples to be tested were prepared into a 10 mg/mL (Dissolve 3 g sample in 300 mL phosphate buffer solution) solution with a phosphate buffer solution of 0.01 mol/L and a pH of 7.6. The solution was homogenized for 1 min to be fully dissolved, centrifuged at 8000 r/min for 20 min, and the protein concentration in the solution was determined by the coomase bright blue method. The supernatant was diluted into different gradients with the same concentration of the phosphate buffer solution. Four millilitres solution of different concentrations was taken and 20 μL 8 mmol/L ANS solution (0.01 mol/L, pH 7.6 phosphate buffer solution was prepared) was added. The fluorescence intensity (FI) in the sample was detected by a fluorescence spectrophotometer within 8–15 min after the shock, and the phosphate buffer solution was used as a blank control. The fluorescence detection conditions were: Excitation wavelength = 390 nm, emission wavelength = 490 nm, and interstitial width = 5 nm. The protein’s relative fluorescence intensity value was the sample’s fluorescence intensity value minus the reagent’s blank value. The protein’s relative fluorescence intensity was used to make the sample concentration curve, and the curve fitting was performed by the least squares method. The initial slope of the line was the surface hydrophobicity index of the sample.

### 2.5. Residual Rate Test

Adhesive samples were placed in an oven and maintained at 120 ± 2 °C until a constant weight (M) was reached. Next, the samples were immersed in water for 6 h in an oven maintained at 60 ± 2 °C, then dried at 105 ± 2 °C for 3 h until a constant weight (m) was reached. The residual rate was calculated as m/M and reported as a percentage.

### 2.6. Three-Ply Plywood Preparation and Evaluation

Three-ply poplar plywood was fabricated under the following production conditions: Coating density (each surface) = 180 g/m^2^, hot pressing time (1.0 MPa) = 5 min, temperature = 120 °C. Plywood samples were stored at room temperature for 12 h after hot pressing.

The wet shear strength was measured in accordance with the Chinese National Standard (GB/T 17657-2013) [21]. Sixteen plywood specimens (25 mm × 100 mm) were cut from as-prepared plywood panels, and plywood specimens were submerged in water for 3 h at 63 ± 2 °C. After the plywood specimens were cooled for 10 min to room temperature, their wet shear strengths were tested using a tensile machine at an operating speed of 10.0 mm/min. The wet shear strengths were calculated using Equation (1) [21]:(1)$$\mathrm{Wet}\;\mathrm{shear}\;\mathrm{strength}\;\left(\mathrm{MPa}\right)\;=\;\frac{\mathrm{Tension}\;\mathrm{Force}\;\left(N\right)}{\mathrm{Bonding}\;\mathrm{Area}\;\left(m^{2}\right)\;}$$

### 2.7. Fourier Transform Infrared (FTIR) Spectroscopy

The adhesive samples were placed in an oven at 120 ± 2 °C to cure completely then ground into ~200 mesh powder in ceramic mortar. The powder was mixed with KBr crystals at a mass ratio of 1:70, and was pressed in a mold to form a sample folium. The FTIR spectra of various powdered adhesives were recorded using a Thermo Nicolet 6700 FT-IR (Nicolet Instrument Corporation, Madison, WI, USA) within a range from 4000 to 400 cm^−1^ at a 4 cm^−1^ resolution, using 32 scans.

### 2.8. Scanning Electron Microscopy (SEM)

For observing the section morphology of the adhesive, the fracture surface of cured adhesive was analyzed using SEM. The adhesive samples were placed in an oven at 120 ± 2 °C to cure completely. The cured adhesive samples, about 1.5 mm in thickness, were fractured and their fracture surface were examined. The fracture surfaces were placed on an aluminum stub, and a 10 nm gold film was coated on using an ion sputter (HITACHI MCIOOO, Ibaraki, Japan). The coated fracture surfaces were observed by a Hitachi S-4800 emission scanning electron microscope (Hitachi Scientific Instruments, Tokyo, Japan).

### 2.9. X-ray Diffraction (XRD)

The adhesive samples were placed in an oven at 120 ± 2 °C to cure completely then ground into ~ 200 mesh powder in ceramic mortar. X-ray diffraction (XRD) patterns were recorded on an XRD diffractometer (XRD-6000, Shimadzu, Kyoto, Japan) using a cobalt source and a 0.2 theta scan ranging from 5° to 60° at 45 kV and 30 mA. The sample determination index was carried out using the program Jade 5.0. The degree of crystallinity is calculated as the ratio of the sum of crystalline band areas to the total area under the XRD patterns [22].

### 2.10. Apparent Viscosity

The apparent viscosity of the adhesive was determined using the rheometer (HAAKE RS1, Waltham, UK) with a parallel plate (35 mm diameter). The distance was set to 1 mm for all measurements. The experiments were conducted under a steady shear flow at 25 °C. Shear rates ranged from 0.1 to 60 s^−1^ in 10 s^−1^ increments. The average value was reported.

### 2.11. Statistical Analysis

All experiments were performed in triplicates. The results were expressed as a mean ± standard deviation and processed using the Origin Pro Version 7.5 software (Origin Lab Corporation, Northampton, UK). The one-way analysis of variance was used for determining the significant difference at *P* < 0.05.

## 3. Results and Discussion

### 3.1. The effects of Denaturing Agent on SPI Adhesive

The purpose of the protein surface hydrophobicity measurement is to indirectly reflect the change of surface hydrophobicity by denaturing agents. The interaction of hydrophobic groups in soybean protein molecules is extremely significant for maintaining the soybean protein’s quaternary structure [23]. Thus, hydrophobic characterization of the protein surface can reflect higher structure changes in the protein. Protein surface hydrophobicity is characterized using the surface hydrophobic index (S_0_). The hydrophobic fluorescent probe (ANS) binds to aromatic amino acids at an excitation wavelength of 390 nm and has a maximum absorption at 470 nm. The fluorescence intensity is proportional to the surface hydrophobicity of the protein [24]. Results for the first set of adhesive samples is presented in Figure 2. The S_0_ of the SPI adhesive is 1577, which is due to being comprised of 28.59% hydrophobic amino acids (tyrosine, phenylalanine, valine, leucine, isoleucine, alanine, methionine) (Table 1). The soybean protein S_0_ increased to reach a maximum of 4572 after urea-denaturation. This increase is explained by the interaction between the urea and protein hydroxyls, which destroys the hydrogen bond, permitting protein molecule unfolding. The addition of SDS increased the protein’s S_0_ to 2787, because SDS destroyed the protein’s structure and caused the hydrophobic groups to evaginate. The addition of SHS caused the protein’s S_0_ to increase, reaching 4082. This phenomenon occurs because SHS destroys the disulfide bond inside the protein molecule, resulting in protein molecule unfolding and the exposure of hydrophobic groups inside protein molecules. At the same time, due to the unfolding of the protein molecule, the hydrophilic group inside the protein is exposed.

The apparent viscosity reflects the fluidity of the protein adhesive. Too high viscosity makes the poor flow quality of the protein adhesive and too low viscosity will make the protein adhesive in the curing process not form an adhesive layer, which will result in a poor bonding performance of the prepared plywood [25]. The apparent viscosity results of the sample with different denaturing agents is shown in Table 3. After adding urea, the initial viscosity of the sample decreases to 48,720 mPa·s, which is because the urea reduces the intermolecular force as a dispersant. The addition of SDS increases the viscosity of the sample to 61,463 mPa·s, because SDS can increase the force between the protein molecules. The addition of SHS sharply reduced the viscosity of samples to 6381 mPa·s, because SHS could hydrolyze protein molecules and reduce the molecular weight of proteins. The pH values of the samples measured in the experiment are all between 6.8 and 7.2, so the change of viscosity is not related to the pH value.

The ATR-FTIR spectroscopy results of the SPI modified with different denaturing agents is shown in Figure 3. Active groups –NH_2_, –COOH, and –OH were identified in the SPI; as were the three characteristic peptide bond peaks –C=O stretching peak at 1627.35 cm^−1^ (amide I) [26], N–H stretching peak at 1515.69 cm^−1^ (amide II) [27], and the C–N stretching peak and N–H bending vibration peak at 1234.68 cm^−1^ (amide III) [28]. The main characteristic peaks’ shape still appears after adding the denaturant, indicating that the soy protein’s molecule primary structure was almost not destroyed. After the addition of urea, amide I, II and III shifted from 1627.35 to 1622.02 cm^−1^, 1515.69 to 1531.19 cm^−1^, and 1234.68 to 1235.20 cm^−1^, respectively. Similarly, the addition of denaturant agents (SDS and SHS) also caused the amide I, II and III to shift. This indicates that adding a denaturant causes the soy protein molecules to stretch and expose more active groups; thus, the protein’s higher structure is destroyed to some extent.

X-ray diffraction patterns and the calculated crystallinity results of the cured adhesive samples are presented in Figure 4. The crystallinity and residual rate can reflect the degree of ordered arrangement of protein molecules. The purpose of the measurement of the residual rate indirectly reflects the degree of modified protein molecule expansion by denaturant agents (Urea, SDS, SHS) and results are showed in Table 3. Figure 4 shows that the SPI samples have two distinct peaks at 9° and 20°, which are the protein molecule’s α-helix structure and β-sheet structure, respectively [29]. After the addition of urea and SDS, the crystallinity and residual rate of the modified adhesive was reduced when compared with the pure protein adhesive. The crystallinity was reduced by 19.36% and 11.26% respectively, and the residual rate was reduced to 83.7% and 82.5% respectively. This is because the urea destroys the hydrogen bond within the protein molecules and SDS evaginated the proteins’ hydrophobic groups, resulting in a destroyed protein molecule and loosely structured protein. The crystallinity of the SHS modified adhesive was 11.73%, which decreased by 42.33% to reach the maximum and residual rate is decreased to 79.4%. On the one hand, SHS destroys the disulfide bond between the protein molecules, and on the other hand, the protein hydrolyzes and its molecular weight decreases. Under such conditions, it is difficult to form an ordered structure, like that of pure proteins, during the curing process.

SEM images of the fracture surface of the cured adhesive are shown in Figure 5. Many circular pores were observed on the fractures surface of the pure protein adhesive, which are easy to be invaded by water and reduce the water-resistance of the SPI adhesive. After adding the urea and SDS denaturation, the fracture surface pores of the adhesive decreases and the surface becomes smooth, compact and continuous, which prevents the invasion of water and improves the water resistance of the adhesive. After the addition of SHS, the fracture surface of the adhesive became smooth and compact, but some small crystals appeared. This may attribute to the aggregation of small molecules hydrolyzed by SHS through hydrogen bonding during the curing process.

The wet shear strengths of the plywood bonded of different denaturing agents (Urea, SDS, SHS) adhesive are shown in Figure 6. The SPI-Urea adhesive due to the fact that urea breaks the hydrogen bond between protein molecules, which makes the protein molecules unfold and exposes the hydrophobic group, resulting in the wet shear strengths of the plywood increasingto 0.88 MPa. Due to the action of SDS, the hydrophobic group in the SPI-SDS adhesive was evaginated, exposing the hydrophobic group within the protein molecule, resulting in the wet shear strengths of the plywood increasingto 0.72 MPa. This is consistent with the surface hydrophobicity analysis. The addition of SHS decreases the wet shear strength of plywood bonded to 0.29 MPa, because SHS can hydrolyze protein molecules and break protein chains, making it difficult to form an effective adhesive layer in the curing process.

To sum up, the improvement of the water resistance of the adhesive and the bonding performance of the prepared plywood are mainly due to the protein molecule unfolding and more hydrophobic groups exposed by the desaturating agents.

### 3.2. The effects of Denaturing Agent on SPI/CA Adhesive

The major functional groups of soy protein molecules include hydroxyl (–OH), carboxyls (–COOH), amines (–NH_2_), and sulfhydryls (–SH) [30]. These groups are associated with side chains of amino acids and terminal groups of proteins. The characteristic peak in CA is the epoxy group, which appeared at 940 cm^−1^. Figure 7 shows the ATR-FTIR spectra of adhesives with CA additive. After the incorporation of CA, its characteristic epoxy group peak disappeared, indicating that CA was cross-linked with active groups on SPI molecules. When compared to the adhesives without CA, it becomes evident that the epoxy group disappeared and a new peak at 1738 cm^−1^ appeared [23], which was related to the carbonyl group of the ester bond and originated from the esterification reaction between epoxy groups and carbonyl groups of SPI. The absorption band at 1396 cm^−1^ was significantly weakened [22], which suggested that the amino group was transformed by the reaction with the epoxy group. The cross-linking structure formed is conductive to improve the water resistance of the adhesive and the wet shear strength of the prepared plywood. The crosslinking mechanism of the soy protein molecule is shown in Figure 8.

SEM images of the fracture surface of the cured adhesive are shown in Figure 9. It can be observed that the fracture surface of the adhesive with CA becomes more compact and continuous than that without CA, which is beneficial to improve the water-resistant bonding performance of the adhesive. Compared with that without CA, a ductile fracture was observed on the fracture surface of the SPI-Urea-CA and SPI-SDS-CA adhesive, which indicated that the generation of the crosslinking structure improved the performance of the adhesive. Compared with that without CA, the fracture surface of the SPI-SHS-CA adhesive becomes more even and uniform, which indicates that the unfolded protein molecules and small molecules generated by the hydrolysis form a uniform cross-linking structure in the hot-pressing process, which is conducive to improving the water-resistant adhesive bonding performance of the adhesive.

The residual rate is used to characterize the degree of cross-linking between protein molecules and results are shown in Table 4. After the addition of CA, the modified adhesives’ residual rates increased to 89.3%, 91.9%, 91.5%, and 90.2%, respectively, and the wet shear strength of the plywood also improved to meet the interior use requirement (≥0.7 MPa) [25]. This improvement came about because the CA epoxy group crosslinked with the amino-, carboxyl-, and other active groups in the protein molecule to form a network structure, which is conducive to improving the adhesive hydrolysis stability, and thus improving the bonding strength of the prepared plywood. The wet shear strengths of the plywood bonded by the modified SPI adhesive with CA is shown in Figure 10. The SPI-Urea-CA, SPI-SDS-CA adhesive wet shear strength was improved by 21.59%, and 45.83% compared with the adhesive without CA added. This is because urea can break the hydrogen bond between the protein molecules to make the protein molecules unfold and SDS can make the protein molecules unfold as a surfactant, so the availability of more hydrophilic groups facilitates cross-linking with CA to form a denser crosslink structure, which resulted in the wet shear strength of the SPI-U-CA (from 0.88 to 1.07 MPa) adhesive and the SPI-SDS-CA adhesive (from 0.72 to 1.05 MPa) increased. SHS broke the disulfide bond in the protein molecule and hydrolyzed protein molecules, allowing the protein molecule chains to unfold and break protein chains. Simultaneously, the hydrophilic groups of the protein molecule chains were cross-linked with CA to improve the wet shear strength of the plywood from 0.29 to 0.92 MPa compared with the SPI-SHS adhesive.

To sum up, the addition of the crosslinking agent (CA) improves the water resistance of soy protein-based adhesives and the bonding performance of the prepared plywood, which is mainly because the active groups on the protein molecules are reacted with epoxy groups to form a crosslinking structure, and has no obvious correlation with hydrophobic groups of soy protein. After the addition of the crosslinking agent, the active groups of the protein molecule unfolded can cross-link with epoxy groups to form a crosslinking structure in the curing process, which improves the water-resistant bonding performance of the prepared plywood.

## 4. Conclusions

The following conclusions are drawn from the experimental results of this study:

(1) The protein molecule hydrogen bond is destroyed under the action of urea and SDS causes the hydrophobic group to evaginate, which increases the surface hydrophobic index of the SPI-Urea adhesive (189.92%) and SPI-SDS adhesive (76.73%) compared with the SPI adhesive. The unfolded protein molecules expose more hydrophobic groups, which improved the bonding performance of the prepared plywood to 0.88 MPa (SPI-Urea adhesive) and 0.72 MPa (SPI-SDS adhesive). The addition of the crosslinking agent CA makes the SPI-Urea adhesive and SPI-SDS adhesive form a denser crosslinking structure during the hot-pressing process and improves the water resistance of the adhesive. Compared with those without CA, the residual rate of the adhesive increased by 9.56% and 10.49% and the wet shear strength increased by 21.59% and 45.83%, respectively. Consistent with the fact that the fracture structure occurs on the fracture surface.

(2) By breaking the disulfide bond between the protein molecules and causing protein hydrolysis, SHS expands and destroys the protein molecular chains to a large extent, so that the surface hydrophobic index increases to 4082. Due to the sharp decline in the initial viscosity of the SPI-SHS adhesive system, it is difficult to form an effective adhesive layer in the curing process, so the wet shear strength of the adhesive decreases to 0.29 MPa. The unfolding of the protein molecular chain and the hydrolyzed small molecule makes it easier for the hydrophilic group to cross-link with the crosslinking agent (CA) and form a cross-linked structure, which improves the degree of cross-linking of protein molecules (the residual rate increased from 79.4% to 90.2%) and the prepared plywood’s bonding strength (the wet shear strength increased from 0.29 to 0.92 MPa).

(3) The modified SPI adhesive with a denaturation agent (urea, SDS, SHS) can improve the bonding performance of the prepared plywood, which is mainly because of improving the water resistance of the adhesive. The addition of the crosslinking agent improves the bonding performance of the prepared plywood, which is mainly due to the formation of the crosslinking structure in the curing process by the reaction between the active groups on the soybean protein molecules and the crosslinking agent.

## Figures and Tables

**Figure 1 polymers-11-01262-f001:**
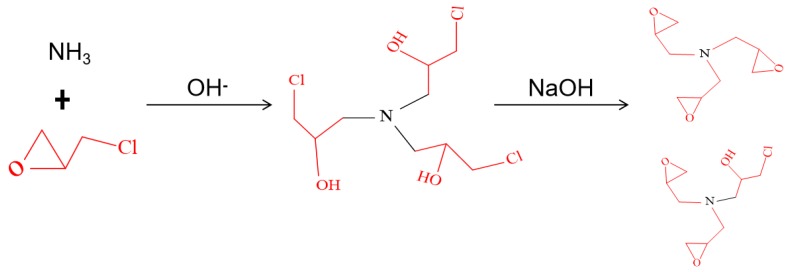
Cross-linking Agent (CA) Synthesis Schematic.

**Figure 2 polymers-11-01262-f002:**
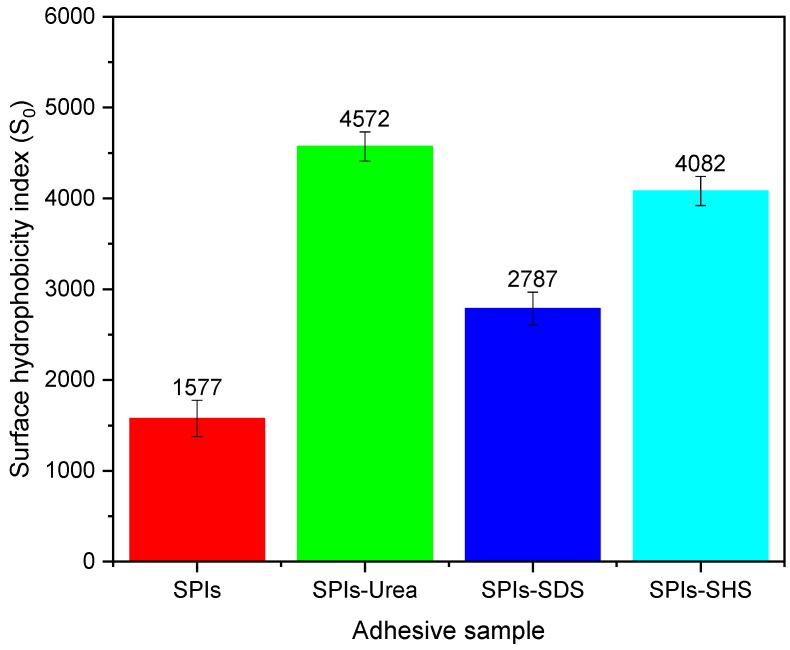
Surface hydrophobicity index of the cured adhesive samples.

**Figure 3 polymers-11-01262-f003:**
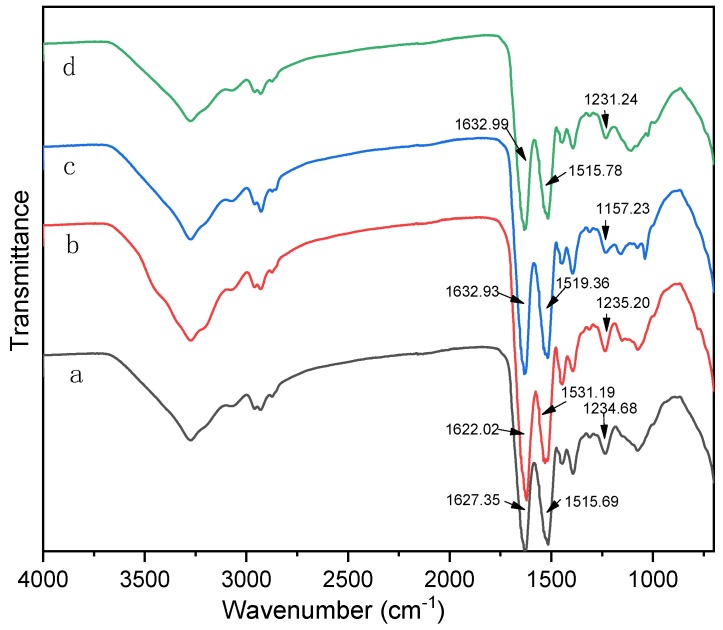
Attenuated total reflection (ATR)-FTIR spectroscopic results from the cured adhesive samples: (**a**) SPI adhesive, (**b**) SPI-Urea adhesive, (**c**) SPI- sodium dodecyl sulfate (SDS) adhesive, (**d**) SPI- sodium hydrogen sulfite (SHS) adhesive.

**Figure 4 polymers-11-01262-f004:**
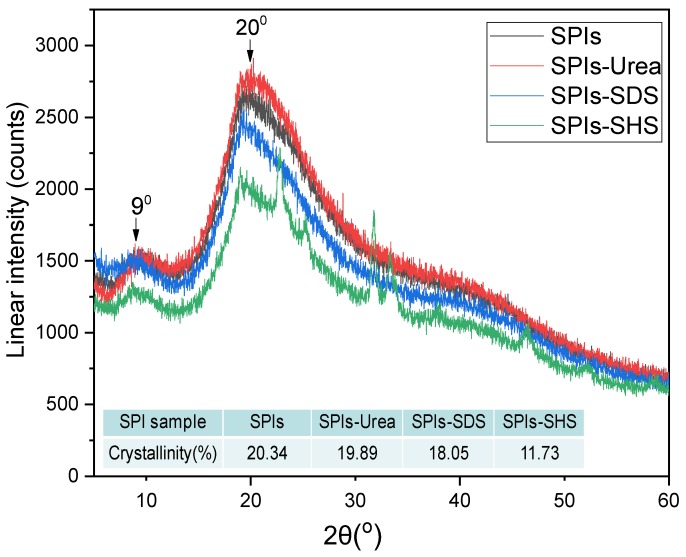
X-ray diffraction pattern and the crystallinity of the cured adhesive samples.

**Figure 5 polymers-11-01262-f005:**
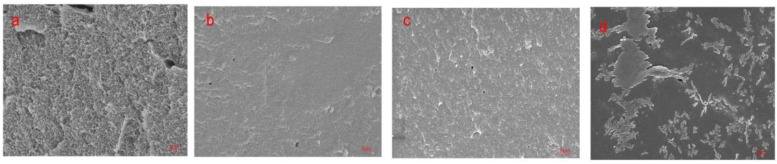
Fracture surface micrographs of the cured adhesive samples: (**a**) SPI adhesive, (**b**) SPI-Urea adhesive, (**c**) SPI-SDS adhesive, (**d**) SPI-SHS adhesive.

**Figure 6 polymers-11-01262-f006:**
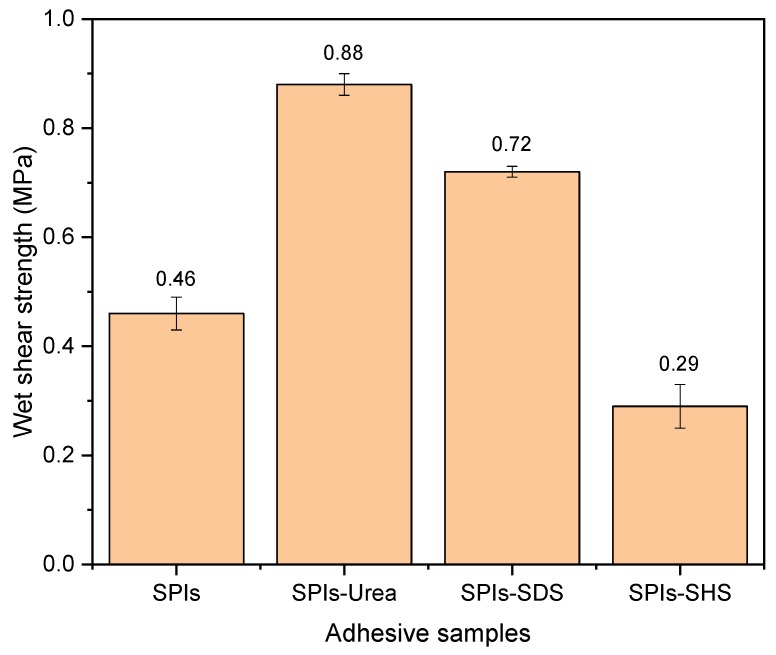
The wet shear strengths of plywood bonded by modified SPI adhesive.

**Figure 7 polymers-11-01262-f007:**
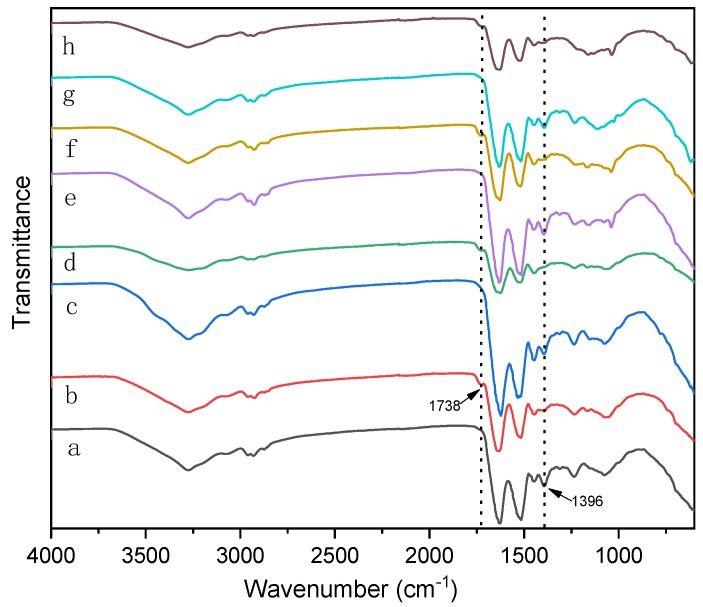
ATR-FTIR spectroscopic of the cured adhesive samples: (**a**) SPI adhesive, (**b**) SPI -CA adhesive, (**c**) SPI-Urea (U) adhesive, (**d**) SPI-Urea-CA adhesive, (**e**) SPI-SDS adhesive, (**f**) SPI-SDS-CA adhesive, (**g**) SPI-SHS-CA adhesive, and (**h**) SPI-SHS-CA adhesive.

**Figure 8 polymers-11-01262-f008:**
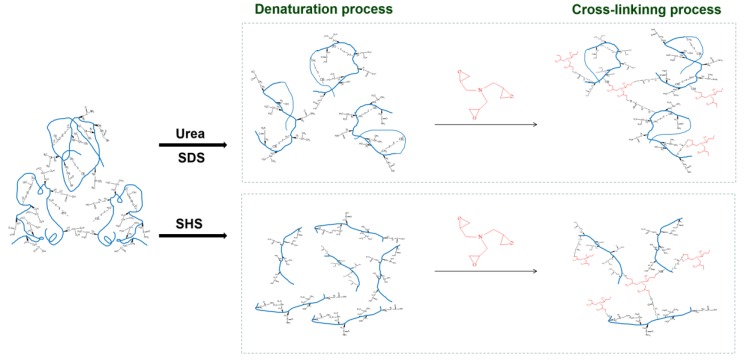
The denaturation and crosslinking mechanism of soy protein molecule.

**Figure 9 polymers-11-01262-f009:**
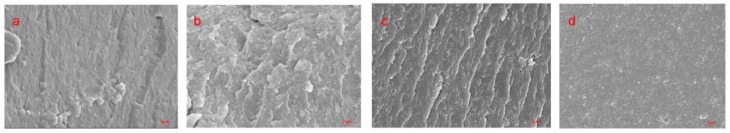
Fracture surface micrographs of the cured adhesive samples: (**a**) SPI-CA adhesive, (**b**) SPI-Urea-CA adhesive, (**c**) SPI-SDS-CA adhesive, (**d**) SPI-SHS-CA adhesive.

**Figure 10 polymers-11-01262-f010:**
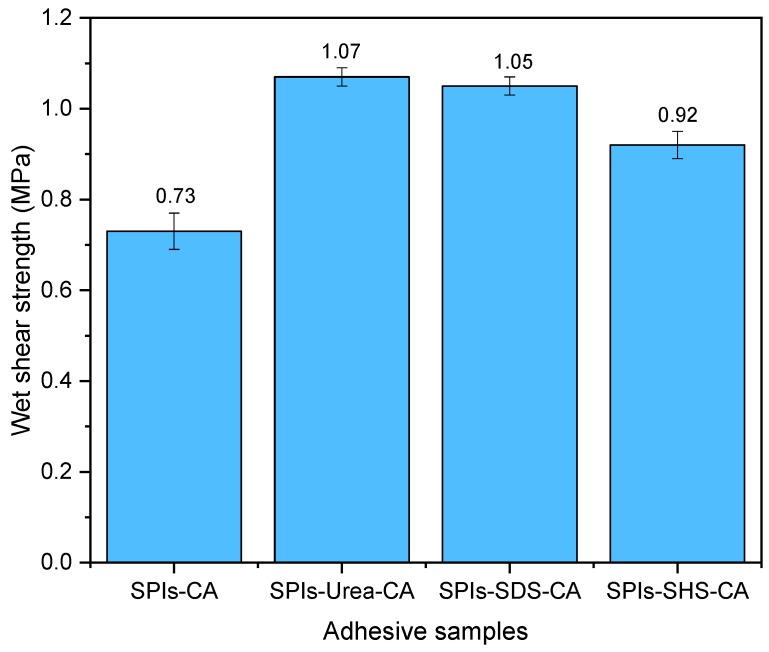
The wet shear strengths of plywood bonded by modified SPI adhesive of with CA.

**Table 1 polymers-11-01262-t001:** The amino acid content of SPI.

Amino Acid Name	SPI
Tyrosine	3.25%
Glycine	3.54%
Leucine	6.81%
Proline	4.54%
Alanine	3.73%
Phenylalanine	4.43%
Valine	5.37%
Isoleucine	3.97%
Serine	4.53%
Threonine	3.33%
Aspartic acid	10%
Glutamic acid	16.65%
Histidine	2.22%
Arginine	6.74%
Lysine	5.37%
Methionine	1.03%

**Table 2 polymers-11-01262-t002:** Formulations used for adhesive samples.

Sample	Formulation			
	SPI	Deionized Water	Denaturing Agents	CA
1 SPI	15 g	85 g		
2 SPI-Urea	15 g	85 g	1 g Urea	
3 SPI-SDS	15 g	85 g	1 g SDS	
4 SPI-SHS	15 g	85 g	1 g SHS	
5 SPI-CA	15 g	85 g		5 g CA
6 SPI-Urea-CA	15 g	85 g	1 g Urea	5 g CA
7 SPI-SDS-CA	15 g	85 g	1 g SDS	5 g CA
8 SPI-SHS-CA	15 g	85 g	1 g SHS	5 g CA

**Table 3 polymers-11-01262-t003:** The residual rate and initial viscosity of the cured adhesive samples.

Samples	SPI	SPI-Urea	SPI-SDS	SPI-SHS
Initial viscosity (mPa·s)	53672	48720	61463	6381
Residual rate (%)	84.8 ± 0.05	83.7 ± 0.02	82.5 ± 0.04	79.4 ± 0.07

**Table 4 polymers-11-01262-t004:** The residual rate of the adhesive: The SPI-CA, SPI-Urea-CA, SPI-SDS-CA, and SPI-SHS-CA adhesive, respectively.

Adhesive	SPI-CA	SPI-Urea-CA	SPI-SDS-CA	SPI-SHS-CA
Residual rate (%)	89.3 ± 0.08	91.9 ± 0.05	91.5 ± 0.07	90.2 ± 0.03

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
