# Peer review of "Effects of Different Denaturants on Properties and Performance of Soy Protein-Based Adhesive"

_polymers, 2019, doi:10.3390/polym11081262_

Round 1

Reviewer 1 Report

I have some comments which are presented below.

1.      Whole point 2.2 should be changed. All the text is from the article: A High Solid Content Bioadhesive Derived from Soybean Meal and Egg White: Preparation and Properties (Luo, J., Li, L., Luo, J. et al. J Polym Environ (2017) 25: 948. https://doi.org/10.1007/s10924-016-0875-3)

2.      Lines 93, 113. Name of Tables should be placed integrated with the tables in the same page.

3.      Line 109. Has Figure 1 been developed by the authors? If not, the authors should refer to the literature.

4.      Lines: 115,125, 126, 127,129, 130, 166, 202, 206, 280. There should be a space between the value and the unit or between different expressions (line 227).

5.      Line 123. In the References the [17] position is different.

6.      Line 142 and others. The authors should use in a lot of fragments of the text the proper abbreviation of “hours”.

7.      Line 147.  What does “ 180 g m-2” mean? Line 153  - is the similar strange expression.

8.      Lines (for example: 142 and 151).  Value and the unit of temperature are presented in the different way. The authors must unify such expressions.

9.      Line 157. Has the equation been developed by the authors? If not, the authors should refer to the literature.

10.  Line 158. The author should use the expression “bonding area” and the unit of this parameter should be presented properly.

11.  Line 160. The authors mentioned:” The cured adhesive sample..”. There is no information about the curing process. The author should  

12.  Lines: 179, 181. The surface hydrophobic index was presented in the different way. What the denotation is proper?

13.  Fig. 2. There is no standard deviation of obtained results in the figure. What was the amount of samples to this test? Or how many repetitions were there during the tests? Why was the analysis not carried out for the remaining samples (Table 2)?

14.  Fig. 4. What does it mean: “2q”?

15.  Fig. 5. There is no unit on the Y axis. The color of the type of columns should be explained in the figure (legend?).

16.  Fig. 7. There is no legend of curves color.

17.  Fig. 8. What does it mean: “temperation”? Some descriptions in figures are illegible. The four figures should be described.

18.  Why there was no statistical analysis of the results obtained, which would give credibility to the presented conclusions?

19.  Line 214. The expression “Table” should be presented in the same way in the whole text.

20.  Line 230. What do the authors mean by this: “van der Waals force”? Is there one kind of force?

21.  Page 6. The text is not justified.

22.  Page 277. What do the authors mean by “-NH2”? and also “(-SH)26.”

       I have suggested the work to be re-written and reviewed. Some fragments of the text were prepared carelessly.

Reviewer 2 Report

The manuscript by Yue et al reports on studies of effect of different denaturants on surface hydrophobicity and crystallinity of denatured protein followed by development of adhesive formulation from denatured protein crosslinked with epoxy-based crosslinked and adhesive strength of resulting formulations. Manuscript has several typos and grammatical mistakes. The explanation offered is very speculative, and in most cases, explanation in one section of the manuscript contradicts with the explanation offered on the other section(s) of the manuscript. I suggest authors to develop the explanations of the results based on the mechanistic approaches of protein disruption with different chaotropic agents studied. The manuscript may be accepted after major revision. Below are some specific comments:

1)      Instead of U, an one-alphabet abbreviation, urea is recommended throughout the text.

2)      Line 67, incomplete sentence. Soy proteins complex quaternary what?

3)      Sentence starting with a number, such as line 129, 4 mL solution of …. Which is not widely accepted in scientific writing. It is recommended to start the sentence with word such as four millilitres solution of….

4)      Line 130, are the numbers correct? Was it 4 mL solution diluted in 20L?

5)      What is residual rate study conducted for? What conclusion did the authors make out of this study?

6)      Shear strength (MPa) is N/mm2. What does percentage (in the denominator) refer to in the formula? It must be the area (in m2) multiplied by 100 or area in mm2.

7)      In line 186 t0 188 authors write “hydrophobic groups in the protein molecule are exposed, and set out to form new subunits through the interaction of different hydrophobic groups. As a result, the hydrophilic groups are exposed and the surface S0 is reduced” inferring that protein denaturation with urea exposes the hydrophilic groups, thus reducing the hydrophobic index (S0) from 2503.1 of unmodified protein to 777.6 of urea denatured protein. Authors propose that the other denaturing agents SDS and SHS lead to unfolding of protein structure, destroyed the protein’s subunit structure, and caused the hydrophobic groups to evaginate evaginating the hydrophobic groups or resulting different subunits due to disulfide cleavage. What are those different sub-units? What is the evidence for existence of such different subunits? What force/molecular interaction makes those sub-units intact? Without any solid experimental evidences, the argument of existence of different sub-units is erroneous. The author’s statement should be supported by experimental evidence.

8)      In lines 195 to 197, authors add that “Because the SHS-denatured protein S0 increased more than that of the SDS denatured protein, it can be surmised that the SHS-denatured protein subunit structure was more loosely bonded and therefore destroyed to a greater extent.” The reviewer’s understanding is that destruction of protein/peptides to a greater extent (as authors claim) would lead to exposure of hydrophilic functional groups as well, which is being overlooked in the manuscript.

9)      In line 207 to 209, authors write “This indicates that adding a denaturant causes the soy protein molecules to stretch and expose more active groups; thus, the protein’s higher structure is destroyed to some extent.” Reviewer’s understanding is that the active functional groups include amine, amide, carboxyl, hydroxyl, etc., which are highly hydrophilic (and thus the denatured protein demonstrates higher water solubility than the native protein). Then, the explanation offered in these lines is in contradiction to the explanation offered in preceding paragraphs where authors say hydrophobic groups are exposed due to denaturation.

10)   In the manuscript, it is repeatedly claimed that “The SDS evaginated the proteins’ hydrophobic groups and the SHS broke the disulfide bond” which appears to be too speculative, but the authors argue this statement with so much confidence without any experimental evidence(s). How was degree of crystallinity measured? How many experiments were conducted? Are the values of degree of crystallinity significantly different for protein treated with different denaturants? I think the authors should develop their argument based on the mechanism of protein disruption with different denaturants. In its form, the explanation is very speculative.

11)   In lines 263to 265, authors write “This is because SHS can destroy the protein subunit to a greater extent by breaking the disulfide bond and exposing more hydrophilic groups”. Then, why the SHS treated protein demonstrated higher surface hydrophobicity (as shown in figure 2)?  

Reviewer 3 Report

The authors present their work on how denaturants can impact crosslinking with triglycidylamine (TGA) and the adhesive function of the resulting material.  The article is well written and generally easy to follow.  Despite this, I am recommending that major revisions are required.  This is mainly because there is no discussion in this manuscript of whether these results represent analysis of triplicate experiments, and/or if the triplicate experiments that may have been performed are experimental or analytical replicates. In most cases, there are no error bars, and statistical analysis is not even described in the materials and methods.  If the results describe experiments that were only performed once, then it is impossible to draw any real conclusions.  Hopefully, all experiments were done in triplicate and therefore statistical can be performed to validate any conclusions drawn.

Aside from this fundamental issue, there are several other questions/comments that I have regarding the manuscript:

1)    The abstract is too long.  The instructions state that the abstract should be 200 words max and this is probably pushing 400.  The text needs to be edited and condensed.

2)    The introduction is missing a key section discussing what has been done in the literature with different denaturants.  The use of denaturants in protein-based adhesive formulations is not novel and the three chosen in this study are commonly used in the field, so specifically with soy protein isolate.  This will help to establish the context and novelty of the research performed here.

3)    Line 98: rmp should be rpm.

4)    Line 101: “Plethora” is not a scientific term that should be used in the materials and methods section of a scientific manuscript.  Is there a volume range that you can report?

5)    Section 2.2: Did the authors do anything to assess their final TGA product?  While the chemistry outlined seems to make sense, it is possible that the reaction did not go to completion and that some of the “TGA” molecules did not actually carry 3 epoxy groups.  Clearly, the effect of TGA in later experiments are evidence that epoxy groups were formed, however, without any characterization of the final TGA product prior to addition, there is great risk that batch to batch variations result in the differences observed in subsequent testing of the crosslinked product.  For instance, was it possible to quantify the number of epoxy groups in the TGA product?

6)    It is good that the authors have used the SPI as a control for their experiments, however, it would have been nice to see a mock treated sample as well, where the SPI was taken through the entire reaction process, but with no denaturant added.  This would help to confirm that the additional 20 minutes at 25 degrees Celcius and the subsequent processing steps did not result in the observed changes.

7)    Line 195-197: What evidence do the authors have that SHS denatures proteins more than SDS?  A 1% SDS concentration appears to be used in these experiments.  This is the same concentration that is commonly used in SDS-PAGE.  The fundamental concept behind SDS-PAGE is that a 1% SDS concentration is capable of linearizing (i.e. denaturing) proteins so that the migration of proteins through the gel can be correlated with their molecular weight.  Alternatively, SHS, which only denatures disulfide bridges, would certainly help open up protein molecules, but hydrogen bonding and other forces would cause the protein to adopt alternative secondary structures.  Even in the mechanism proposed by the authors in Figure 6, the pathways using SDS and SHS do not appear to be that different, with regards to the  opening up of the secondary structure, and accessibility to TGA.

8)    I am confused with the authors’ ideas regarding the secondary structures of their materials and their conclusions.  In Lines 204-205. The authors state, “The peak shape did not change after adding the denaturant, indicating that the soy protein’s molecule secondary structure was almost destroyed.”  Then in Lines 228-229, the authors state, “Adding the denaturant did not change the crystallinity peak, indicating that the protein molecule’s secondary structure remained unchanged.”  The two arguments are contrary to one another.

9)    Line 226: Figure 4, not Figure 5?

10) Figure 5: Not clear what orange and green represent.

11) Figure 7: Not labelled.

12) Line 279-280: Very difficult to compare one FTIR spectrogram to another without an overlay (i.e. -/+ TGA).  That being said, my understanding is that the primary amine group is in the 3300-3400 range, not the 1300 range.  The 1300 region may indicate the C-N stretch of amine (aromatic).  Even still, I am not convinced that the peak at 1304 disappears.  Would be best to point to the exact peak that you are arguing disappears, and it would be nice to directly compare this peak in samples +/- TGA.

13) Line 299: The other state that, “the epoxy crosslinking agent formed a crosslinked structure with some unstable new structures that were simultaneously generated.”  It is not clear to me what this means, nor what evidence that there would be from TGA that would allow for such a conclusion. 

14) Line 300-303: The authors write that, “in the third stage, the degradation peak of the modified adhesive with TGA decreases and moves towards higher temperature, indicating that the soy protein molecules crosslinked with TGA forming a more stable, cross-linked structure and improving thermal stability.”  The graphs observed in the 4 panels of Figure 8 do not tell the same story, and thus this sentence is an over-simplification of the data.  In addition, the authors claim that in stage 3, the crosslinked molecules are more stable, but in the two panels on the left, even at 400 degrees, there appears to be more deterioration in the +TGA samples, which would indicate less stability.  In the two panels on the right, there is very little difference between the +/- TGA samples.  Again, it is all moot if the experiments were not performed in triplicate and if statistical analysis cannot be performed.

Round 2

Reviewer 2 Report

The revised manuscript has been greatly improved. Nevertheless, the added sections (those in red) suffer from poor sentence structure and grammatical errors, thus the entire manuscript requires English polishing. Even though the authors followed the method that has been peer reviewed and published elsewhere for surface hydrophobicity measurement, I am not still convinced with the type of experiment conducted, the results obtained, and the explanation offered regarding surface hydrophobicity of developed adhesive formulations. I would recommend re-writing/re-structuring this section in a more convincing way.

One specific comment: in line 20 (and afterwards), does the abbreviation CA stand for crosslinking agent? It is recommended to use the abbreviation for the name of the reagent triglycidyl amine (maybe TGA or whatever sounds suitable to authors) rather than CA.

Reviewer 3 Report

The authors present a much better manuscript that is a significant improvement on the previous draft.  There are still several grammatical errors throughout the entire manuscript.  I would highly recommend having someone review before final submission. (i.e. Line 107: "feather" instead of "features", Line 16-18: Sentence is not complete, or not worded properly.).